

**Efficient surrogate modeling methods for large-scale Earth system models based on**
**machine learning techniques**
**Dan Lu[1,*], Daniel Ricciuto[2]**
[1]Computational Sciences and Engineering Division, Climate Change Science Institute, Oak
Ridge National Laboratory, Oak Ridge, TN, USA;
[2]Environmental Sciences Division, Climate Change Science Institute, Oak Ridge National
Laboratory, Oak Ridge, TN, USA;
* Corresponding Author: Dan Lu, lud1@ornl.gov

19                              December 2018

20              For Publication in Geoscientific Model Development





## Abstract

Improving predictive understanding of Earth system variability and change requires data-model

integration. Efficient data-model integration for complex models requires surrogate modeling to

reduce model evaluation time. However, building a surrogate of a large-scale Earth system

model (ESM) with many output variables is computationally intensive because it involves a large

number of expensive ESM simulations. In this effort, we propose an efficient surrogate method

capable of using a few ESM runs to build an accurate and fast-to-evaluate surrogate system of

model outputs over large spatial and temporal domains. We first use singular value

decomposition to reduce the output dimensions, and then use Bayesian optimization techniques

to generate an accurate neural network surrogate model based on limited ESM simulation

samples. Our machine learning based surrogate methods can build and evaluate a large surrogate

system of many variables quickly. Thus, whenever the quantities of interest change such as a

different objective function, a new site, and a longer simulation time, we can simply extract the

information of interest from the surrogate system without rebuilding new surrogates, which

significantly saves computational efforts. We apply the proposed method to a regional ecosystem

model to approximate the relationship between 8 model parameters and 42660 carbon flux

outputs. Results indicate that using only 20 model simulations, we can build an accurate

surrogate system of the 42660 variables, where the consistency between the surrogate prediction

and actual model simulation is 0.93 and the mean squared error is 0.02. This highly-accurate and

fast-to-evaluate surrogate system will greatly enhance the computational efficiency in data-

model integration to improve predictions and advance our understanding of the Earth system.



## 1 Introduction

Improving predictive understanding of Earth system variability and change requires data-model integration. For example, Bilionis et al. (2015) improved Community Land Model (CLM) prediction of crop productivity after model calibration; Müller et al. (2015) improved the CLM prediction of methane emission after parameter optimization; and Fox et al. (2009) and Lu et al. (2017) improved the terrestrial ecosystem model predictive credibility of carbon fluxes after uncertainty quantification. However, data-model integration methods are usually computationally expensive involving a large ensemble of model simulations, which prohibits their application to complex Earth system models (ESMs) with lengthy simulation time. To reduce computational costs, surrogate modeling is widely used (Razavi et al., 2012; Gong et al, 2015; Ray et al., 2015; Huang et al., 2016, Lu et al., 2018; Ricciuto et al., 2018). The surrogate model, which is a set of mathematical functions, approximates the actual simulation model based on pairs of simulation model input-output samples, and then replaces the simulation model in the data-model integration. As the ESMs evaluation is expensive, it is desired to use a limited number of ESM simulation samples to build an accurate surrogate. As the surrogate model needs to be calculated many times in data-model integration, it is required to build a fast-to-evaluate surrogate. In this study, we use a very few simulation model runs to build an accurate and fast evaluated surrogate system of a large scale problem based on advanced machine learning methods.

In Earth system modeling, we usually need to build a surrogate system of many output variables over large spatial and temporal domains. ESMs tend to be simulated in a regional or global scale with many grid cells for several years, producing a large number of output variables. In addition, ESMs are used to solve versatile scientific problems, so the quantities of interest



(QoIs) often change. Moreover, the development of a surrogate requires expensive ESM runs,
and a large number of runs are often needed to capture the complex model input-output
relationship. Therefore, it is reasonable to build a surrogate system for all possible model outputs
to reduce the efforts of rerunning ESMs for a new surrogate development when the QoIs change.
In this way, whenever we simulate the outputs in a new site or for additional sites, at a different
time or for a longer period, we can simply extract the information of interest from the large
surrogate system without spending extra efforts in building new surrogates, which significantly
saves the computational costs.

Building and evaluating a surrogate system of a large number of model outputs can be very

computationally intensive for almost all the surrogate methods. Polynomials and artificial neural
networks are widely used for surrogate modeling (Razavi et al., 2012; Viana et al., 2014).
Polynomial methods, such as polynomial regression and radial basis functions, need to solve
polynomial coefficients in the surrogate construction and to calculate matrix multiplications in
the surrogate evaluation. Using a $p$th-order polynomial to approximate a model with $d$
parameters, $M = (p+d)!/(p!d!)$ coefficients need to be solved, i.e., the number of coefficients
increases factorially fast with the parameter size and polynomial order. When $d$=40, a second-
order polynomial involves 861 coefficients and a third-order polynomial involves 12341
coefficients. ESMs have many uncertain parameters and a high-order polynomial is usually
needed to approximate complex ESMs, which can easily lead to a prohibitive number of model
evaluations, up to ~$10^5$, necessary to compute the polynomial coefficients. To reduce the
computational costs, some regularization techniques such as Bayesian compressive sensing have
been used (Sargsyan et al., 2014; Ricciuto et al., 2018). These regularization techniques can use a
few samples to solve a large number of coefficients (i.e., an underdetermined system) by





iteratively minimizing the L1 norm of the coefficient vector. But they usually perform
minimization once for one model output, so for a large model outputs problem, significant
computing effort is required. To reduce the computing burden in building polynomial-based
surrogates, we need to reduce the output dimensions.

Reducing the model output dimensions also improves computational efficiency in the

evaluation of the polynomial-based surrogates. For example, evaluating the third-order
polynomial-based surrogate of the model with 40 parameters and 300,000 outputs at 1 parameter
sample, we need to calculate two matrix multiplications where matrix A has the size [1, $M$] and
B has the size [$M$, Nout] and $M$ =12341 and Nout=300,000. The surrogate evaluation takes about
90 seconds and most time is spent on loading the huge matrix. When Nout reduces to 20, the
surrogate evaluation quickly reduces to less than a second. Note that an ESM can easily have
more than 40 parameters and more than 300,000 model outputs. Even using the most advanced
supercomputers with GPUs, the data storage and loading are still a bottleneck. Thus, reducing
model output dimensions is necessary for both fast building and evaluating polynomial-based
surrogates.

Neural network (NN) assisted surrogate modeling also suffers from high computational

costs when applied to a large-scale problem with many QoIs. To approximate a complex ESM
with many outputs, a complicated NN with many wide hidden layers is usually needed to capture
the complex relationship between the model inputs and outputs, because each spatial and
temporal output variable is driven by different meteorological forcing such as air temperature,
humidity, wind speed, precipitation, and radiation. The full connections between nodes in the
input layer and the first hidden layer, between nodes of the hidden layers, and between nodes in
the last hidden layer and a large number of nodes on the output layer, involve a great amount of



NN weights and biases that need to be solved. For the same example discussed above, to
approximate the model with 40 parameters and 300,000 model outputs, an NN with two hidden
layers and each layer having 100 nodes has over 30 million weights and biases. Calculation of
these weights and biases requires many samples to train the NN for a good fit. Each training
sample involves one model evaluation. However, ESM simulation is time consuming, which
usually takes several hours or days and can be up to months or even years. A limited sample size
is not enough to train a deep and wide NN for convergence and a simple NN trained by a small
sample size may not capture underlying Earth systems accurately. Thus, reducing model output
dimensions is needed to advance the NN-based surrogate modeling. A small output size reduces
the width of the output layer and also simplifies the relationship between the model inputs and
outputs, so that a simple NN architecture can be appropriate and a small sample size can be
sufficient to accurately train the simple NN. In addition, a simple NN can also be fast evaluated
with small weight matrix multiplications.

In this work, we propose to use singular value decomposition (SVD) to reduce model

output dimensions, so as to improve the computational efficiency in both building and evaluating
the surrogates. ESM outputs usually show periodic changes along time and strong correlations
between locations, which promises a fast decay of singular values. So, we can use a small
number of singular value coefficients to capture a great amount of output information, enabling a
significant output dimension reduction. We use the NN for surrogate modeling, because
compared to polynomial methods, NNs have shown less difficulty in fitting highly nonlinear and
discontinuous functions which are usually observed in ESMs response surfaces. For example,
carbon flux state variables, such as gross primary productivity (GPP), are strongly affected by
vegetation related parameters. When the parameter samples cause zero vegetation growth, GPP



has zero values. Whereas when the parameter samples cause high vegetation growth, GPP has
large positive values. This leads to a discontinuous GPP response surface jumping from zeros to
nonzeros.

NNs theoretically can fit any functions, but their practical performance strongly depends

on the NN's architectures and hyperparameters. NN has many hyperparameters such as the
number of layers, number of nodes in each layer, type of activation functions, and learning rate
of the stochastic gradient descent optimization. A slight change in the hyperparameter value can
result in dramatically different NN performance. Development of a high-performing NN is time-
intensive and usually requires trial-and-error tuning by machine learning experts. In this work,
we use Bayesian optimization techniques to optimize the NN architecture and hyperparameters
so as to produce an accurate NN model for the training data. Bayesian optimization searches the
hyperparameter space to iteratively minimize the validation errors of the NN by balancing
exploration and exploitation (Shahriari et al., 2016). Researches suggested that Bayesian
hyperparameter optimization of NNs is more efficient than manual, random, or grid search with
better overall performance on test data and less time required for optimization (Bergstra et al.,
2011; Snoek et al., 2012). Bayesian optimization involves a large ensemble of NN fittings and it
is a sequential model-based optimization, thus, fast training of the NN models is important. Our
proposed SVD method can simplify the NN architecture so as to advance the NN training and
improve the Bayesian optimization performance.

In this effort, we propose an SVD-enhanced, Bayesian-optimized, and NN-based surrogate

method and aim to build an accurate and fast-to-evaluate surrogate system of a large-scale model
using a few model runs, so as to improve computational efficiency in surrogate modeling and
thus advance the data-model integration. We apply the method to a simplified land model in the



Energy Exascale Earth System Model (sELM) to improve the model predictive capability of
carbon fluxes. We build a surrogate system of 42660 model output variables which are annual
GPPs at 1422 locations simulated for 30 years. The sELM is a regional-scale terrestrial
ecosystem model that simulates terrestrial water, energy, and biogeochemical processes in
terrestrial surfaces. Simulation of sELM is important for improving our understanding of
ecosystem responses to climate change. However, sELM requires lengthy times for hydrologic
and carbon cycle equilibration, and these high computational costs limit the affordable number of
simulations in data-model integration thus resulting in poor model performance. The proposed
machine learning assisted surrogate method makes the sophisticated data-model integration
computationally feasible and promises an improvement of the sELM predictions.

The major contributions of this work are (1) using SVD to reduce model output

dimensions so as to improve computational efficiency in both building and evaluating an
accurate surrogate of a large-scale ESM; (2) using Bayesian optimization techniques to fast
generate an accurate NN-based surrogate; and (3) applying the proposed method to build a large
surrogate system of a regional-scale ESM to advance data-model integration. To our knowledge,
the method of using SVD to enhance surrogate modeling is novel and we have not seen the
application of Bayesian optimization to improve NN-based surrogates in Earth system modeling.

The paper is organized as follows. In section 2, we first describe the sELM, the model

parameters and the QoIs we build surrogates for; following that, we introduce the SVD, NNs,
and Bayesian optimization methods. In section 3, we apply the methods to the sELM and analyze
the surrogate accuracy. In section 4, we discuss strategies to improve surrogate accuracy and
investigate our method's performance in the application of these strategies. In section 5, we end
this paper by drawing our conclusions.



## 2 Materials and Methods

### 2.1 Description of sELM and related parameters



We developed a simplified version of Energy Exascale Earth System (E3SM) land model
(ELM), or sELM, to simulate carbon cycle processes relevant for Earth system models in a
computationally efficient framework. This framework allows us to perform large regional
ensembles that are computationally infeasible using offline land surface models such as ELM.
sELM is a combination of model elements from the Data Assimilation Linked Ecosystem
Carbon model (DALEC; Williams et al., 2005) and the Community Land Model version 4.5
(CLM4.5; Oleson et al., 2013). sELM consists of five process-based submodels that simulate
carbon fluxes between five major carbon pools using 49 overall parameters. Based on previous
sensitivity analysis using ELM (Ricciuto et al., 2018), this study considers the most sensitive
eight parameters associated with four out of the five submodels. We summarize all five process-
based submodels and their interactions below and in Figure 1.
sELM consists of five major submodels: photosynthesis, autotrophic respiration,
allocation, deciduous phenology, and decomposition. Photosynthesis is driven by the aggregate
canopy model (ACM) from the DALEC, which itself is calibrated against the soil-plant-
atmosphere model (Williams et al., 2005). ACM predicts GPP as a function of carbon dioxide
concentration, leaf area index, maximum and minimum daily temperature, and
photosynthetically active radiation.  Here the GPP predicted by ACM is modified by BTRAN,
which reduces GPP when soil water is insufficient to support transpiration. Because sELM does
not predict soil moisture, BTRAN is calculated in a full ELM simulation and is fed into sELM as
an input. ACM shares one parameter, the leaf carbon to nitrogen ratio (*leaf C:N*), with the





autotrophic respiration model and employs an additional parameter, the specific leaf area at the
top of the canopy (*slatop*).

The remaining four submodules are based on ELM. The autotrophic respiration model

computes the growth and maintenance respiration components and is controlled by four
parameters, the *leaf C:N*, the fine root carbon to nitrogen ratio (*froot C:N*), the base rate of
maintenance respiration (*br_mr*), and temperature sensitivity for maintenance respiration
(*q10_mr*). The allocation model partitions carbon to several vegetation carbon pools following
those in ELM: leaves, fine roots, live stem, dead stem, live coarse roots and dead coarse roots. In
the allocation model, we only consider one parameter, the fine root to leaf allocation ratio
(*froot_leaf*). The deciduous phenology model is used to predict the timing of budbreak and
senescence. It considers two parameters, the critical day length to initiate autumn senescence
(*crit_dayl*) and the number of accumulated growing degree days needed to initiate spring leaf-out
(*crit_onset_gdd*). The last submodel is a decomposition model that simulates heterotrophic
respiration and the decomposition of litter into soil organic matter using the converging trophic
cascade framework as in the CLM4.5 (Oleson et al., 2013). Because this study focuses on plant
carbon uptake, no uncertain parameters are considered in the decomposition model. In sELM,
nutrient feedbacks are not represented explicitly, however a constant nitrogen limitation factor is
included to downregulate photosynthetic uptake.

The sELM can simulate several carbon state and flux variables as shown in Figure 1 with

green shapes. GPP, which represents the total plant carbon uptake, is considered in this study.
Here we use sELM to predict annual GPP in deciduous forest systems in the eastern region of the
United States for 30 years between 1981-2010. The carbon state variables are spun up to steady
state by cycling the GSWP3 input meteorology (Kim et al., 2017) from 1981-2010 for 5 cycles,



and the 6th cycle is used as the output for our surrogate modeling study. The region of interest
covers 1422 land grid cells (locations) as shown in Figure 2. Given 30 outputs at each location
(annual values over 30 years), a total of 42660 GPP variables are simulated. The model uses one
plant functional type and the phenological drivers such as air temperature, solar radiation, vapor
pressure deficit, and $CO_2$ concentration are used as boundary conditions. One regional sELM run
takes about 24 hours on a single processor, which although much faster than ELM is still
computationally too expensive to be directly used in model-data integration studies. To improve
the computational efficiency in generating the sELM simulation samples to develop the surrogate
model, we use high performance computing to perform an ensemble of 2000 sELM model
simulations in parallel. The 2000 parameter input samples are randomly drawn from the
parameter space defined in Figure 3. The numerical ranges of these parameters are designed to
reflect their average values and broad uncertainties associated with the temperate deciduous
forest plant functional type. The output samples are sELM simulated GPPs at the 1422 locations
for 30 years. In the surrogate modeling, part of the 2000 input-output samples are used for
developing the surrogate and part of them are used to evaluate the surrogate accuracy, as
discussed in section 3.
**2.2   Efficient surrogate modeling methods**

In this section, we introduce our SVD-enhanced, Bayesian-optimized, and NN-based

surrogate methods. We first describe the SVD for reducing data dimensionality, then introduce
the NN techniques for building a surrogate model, and last depict the Bayesian optimization
algorithm for producing a high-performing NN-based surrogate.



### 2.2.1 Singular value decomposition for data compression


We build a surrogate system of model outputs by fitting a data matrix whose columns are
output variables and rows are output samples. For a model with 100000 output variables, the
columns of this matrix span a 100000-dimensional space. Encoding this matrix on a computer
takes quite a lot of memory and evaluating this matrix takes a large number of calculations. We
are interested in approximating this matrix with some low-rank matrix but remaining its most
information, so as to reduce data transfer and accelerate matrix calculation.
Singular value decomposition (SVD) decomposes a matrix $\mathbf{A}$ with size $m \times n$ into three
other matrices, $\mathbf{A} = \mathbf{U}\mathbf{S}\mathbf{V}^T$, where $\mathbf{U}$ is an $m \times m$ orthogonal matrix, $\mathbf{V}$ is an $n \times n$ orthogonal
matrix, and $\mathbf{S}$ is an $m \times n$ diagonal matrix saving singular values in descending order on the
diagonal. Truncated SVD keeps the $K$ largest singular values and corresponding $K$ column
vectors of $\mathbf{U}$ and $K$ row vectors of $\mathbf{V}^T$ to form $\widetilde{\mathbf{A}} = \mathbf{U}_K \mathbf{S}_K \mathbf{V}_K^T$. The $K$-rank matrix $\widetilde{\mathbf{A}}$ has proven to
be the best approximation of $\mathbf{A}$ in minimizing the Frobenius norm of the difference between $\mathbf{A}$
and $\widetilde{\mathbf{A}}$ under the constraint of rank$(\widetilde{\mathbf{A}}) = K$. In addition, the total of the first $K$ singular values
divided by the sum of all the singular values is the percentage of information that those singular
values contain. For example, if we want to keep 90% of the data information, we just need to
compute sums of $K$ largest singular values until we reach 90% of the sum and discard the rest.
By dropping all but a few singular values and then recomputing the approximated matrix, the
SVD technique compresses the data information and reduces data dimensions. When the matrix
$\mathbf{A}$ shows strong correlations between columns (variables), a low-rank matrix $\widetilde{\mathbf{A}}$ can make a very
accurate approximation of $\mathbf{A}$.
In this study, we use SVD to reduce training data dimensions. The training data matrix $\mathbf{A}$
$[m, n]$ for surrogate construction contains model output samples information. $n$ columns are



output variables (e.g., the 42660 temporal and spatial GPPs in this work) and $m$ rows are the
samples of these variables (e.g., the sELM simulation results of the 42660 GPPs for the $m$
parameter samples), and usually $n \gg m$ for expensive ESMs with many outputs. In
implementation, we first perform truncated SVD to get low-rank matrices $\mathbf{U}_K[m, K]$, $\mathbf{S}_K[K, K]$,
and $\mathbf{V}_K^T[K, n]$ with $K \ll n$, we then use the low-dimensional dataset $(\mathbf{V}_K^T \mathbf{A}^T)^T$ with reduced size $m$
$\times K$ as training data to build the surrogate model of the $K$ largest singular value coefficients.
Next, we evaluate the surrogate model at $q$ new data points to get results $\mathbf{Y}_{new}$ with size $q \times K$.
Lastly, we transform the predicted values back to its original size $q \times n$ through $\mathbf{Y}_{new} \mathbf{V}_K^T$ to obtain
the surrogate approximation of the $n$ variables at the $q$ new data points.
**2.2.2   Neural networks for surrogate modeling**
Artificial neural networks (NNs) consist of fully connected hierarchical layers with nodes
which can be flexibly used for function approximation (Yegnanarayana, 2009). The first layer is
the input layer and each node in the input layer represents one model input variable. The last
layer is the output layer and each node in the output layer represents one model output variable.
The layers between input and output layers are hidden layers which are used to approximate the
relationship between model inputs and outputs. When the relationship is complex, a complicated
NN with many wide hidden layers is usually needed. The input layer first assigns model
parameter values to its nodes. Then each node in the first hidden layer takes multiple weighted
inputs, applies the activation function to the summation of these inputs, and calculates the node's
value. Next, the second hidden layer takes the values on the first hidden layer nodes as inputs
and calculates its nodes' values in the same way. This process moves forward till we get values
of all nodes in the output layer, i.e., obtaining NN predictions for the given model parameter
input values. The nodes in each layer are fully connected to all the nodes in its previous and





subsequent layers. Each of these connections has an associated weight and bias. A complicated
NN results in a large number of weights. By tuning these weights and biases based on some
training data, we improve the NN approximation of the underlying simulation model.

NN uses stochastic gradient descent (SGD) method to optimize its weights and biases

(Bottou, 2012). SGD optimizes variables by minimizing some loss function based on the
function's gradients to these variables. The loss function is usually defined as the mean squared
error (MSE) between the NN predictions and model simulations for the same set of model
parameter samples in the training data. SGD iteratively updates the optimized variables at the
end of each training epoch. In the process, the learning rate, which specifies how aggressively
the optimization algorithm jumps between iterations, greatly affects the algorithm's performance
and has to be tuned. A small learning rate will take a long time to reach the optimum causing a
slow convergence, whereas a big learning rate will bounce around the optimum causing unstable
results and a difficult convergence. Using SGD to optimize a complex NN with many weights
requires a great amount of computational efforts and has difficulty in convergence. First, many
training data are required to tune a large number of weights. Small training data can easily cause
over-fitting, i.e., the NN "perfectly" fits the training data but performs badly on new data, thus
deteriorating the NN prediction accuracy. In addition, a large number of weights involve massive
matrix calculations in evaluating the loss function, slowing down the training process.
Furthermore, a complicated NN has difficulty in convergence and can easily get stuck in local
minima. In this work, we use SVD to reduce the model output dimensions, so as to decrease the
number of nodes in the output layer and simplify the NN architecture, thus reducing the size of
the weights and enabling a reasonable NN training from small training data, and ultimately
improving the computational efficiency.



### 2.2.3 Bayesian optimization algorithm for NN hyperparameter optimization


NN involves a lot of hyperparameters that dramatically affect its performance such as the
number of layers, the number of nodes in each layer, and the learning rate of the SGD algorithm.
Hyperparameter optimization is needed to produce a high-performing NN. This requires
optimizing an objective function $f(x)$ over a tree-structured configuration spaces $x \in X$, where
some leaf variables (e.g., the number of nodes in the third hidden layer of an NN) are only well
defined when branch variables (e.g., a discrete choice of how many layers to use) take particular
values. In addition, the optimization not only optimizes discrete and continuous variables, but
also simultaneously choose which variables to optimize. When the NN is used for surrogate
modeling, the objective function is the NN accuracy of predicting some validation data. In this
case, the $f(x)$ does not have a simple closed form but can be evaluated at any arbitrary query
point $x$ in the configuration space. For such optimization problem, a sequential search method is
needed, besides some inefficient grid search and random search approaches (Bergstra and
Bengio, 2012). The sequential search method starts with some random points in the search space,
and then iteratively evaluates new points based on NN predictions on previously evaluated
points. After $N$ evaluations, we choose the optimal combination of the hyperparameters resulting
in the highest NN prediction accuracy. Among the sequential search algorithms, Bayesian
optimization is able to take advantage of full information provided by the history of the
optimization to improve the search efficiency.
Tree-structured Parzen estimator (TPE) and Gaussian process are two widely used
Bayesian optimization algorithms (Shahriari et al., 2016; Bardenet and Kegl, 2010; Niranjan et
al., 2010; Snoek et al., 2012). In comparison to the Gaussian process, TPE works well for all
types of NN hyperparameter variables, is robust to NN randomization, has a fast calculation and



a straightforward implementation without associated hyperparameters specification (Bergstra et
al., 2011). In this work, we use the TPE algorithm for NN hyperparameter optimization.
**3   Results**

In this section, we present the results of building the surrogate system of 42660 GPP

variables of sELM. First, we demonstrate that our method using SVD can efficiently build and
evaluate a large surrogate system by comparing the results with and without application of SVD.
We then investigate the influence of NN's architecture on surrogate performance and show that
our method using hyperparameter optimization can fast generate an accurate NN. Last, we
evaluate surrogate accuracy on the large-scale spatial and temporal GPPs.

We consider three sets of data, the training data for fitting the NN, the validation data to

detect overfitting in the NN training and to select the best-performing NN in the hyperparameter
optimization, and the test data to evaluate the NN prediction accuracy. Each data set contains
pairs of parameter and GPP samples. The parameter samples are randomly drawn from the
parameter space defined in Figure 3. To assess the effectiveness of our proposed surrogate
method for a small data set, we consider only 20 training data (Figure 3). The validation data is
chosen as 0.3 fractions of the training data. The NN model will not train on the validation data
but evaluate the loss function on them at the end of each epoch. In each epoch, the training data
is shuffled, and the validation data are always selected from the last 0.3 fraction. Precisely, we
only use 14 samples to tune NN weights. Attribute to shuffling, these 14 samples can be a
different subset from the 20 training data in each epoch, thus we sufficiently explore the limited
20 data information for building the surrogates. We use 1000 test data (Figure 3) to evaluate the
NN prediction accuracy, which makes a reasonable assessment of our proposed method within
an affordable computational cost. Note that the 1000 test data are not needed for building the





surrogates but used to demonstrate the effectiveness and efficiency of our method. When using
our method to build the surrogates of the 42660 GPPs, only 20 sELM model simulations are
used.

We define the loss function as the mean squared error (MSE) between the NN predictions

and the sELM simulations based on the parameter samples for training. We use Adam algorithm
(Kingma and Ba, 2015) for stochastic optimization of NN and run it for 800 epochs to minimize
the loss function and update NN weights. Adam has been shown a superior stochastic
optimization algorithm in training NN (Basu et al., 2018). There is no right answer for the
optimal number of epochs. A small number of epochs could result in underfitting and a large
number of epochs may lead to overfitting. Here we consider a large number of epochs and in the
meantime use early stopping to avoid overfitting. During the training, when there is no
improvement of loss functions for the validation data in 100 epochs, we stop the training and
choose the weights at the epoch resulting in the smallest loss function of the validation data as
the optimal weights and the associated NN as the best trained NN under the given setting.

We then use the trained NN to predict the 1000 test data and compare the predictions with

the corresponding sELM simulation results to evaluate the NN accuracy. We define two metrics
for evaluation, the MSE and the coefficient of determination. The MSE computes the expected
value of the squared prediction errors; the small the MSE value is, the better the prediction. The
coefficient of determination, also called $R^2$ score, measures how well the unobserved data are
likely to be predicted by the NN model. Denote $\hat{y}_i$ as the NN prediction of the $ith$ sample and $y_i$
as the corresponding sELM simulation, the $R^2$ score estimated over $N_s$ samples is defined as
$R^2 = 1 - \frac{\sum_{i=1}^{N_s}(y_i - \hat{y}_i)^2}{\sum_{i=1}^{N_s}(y_i - \bar{y}_i)^2}$, where $\bar{y} = \frac{1}{N_s}\sum_{i=1}^{N_s} y_i$. Best possible value of $R^2$ score is 1.0, indicating
that the NN can perfectly predict the test data. $R^2$ score can be negative indicating the model is





arbitrarily poor. A constant model gets a $R^2$ score of 0.0. Compared to MSE, the $R^2$ score
considers the variability of the data which provides a more reasonable measure.
**3.1   SVD reduces data dimensionality and improves surrogate efficiency**
We consider two scenarios when building the surrogate system of the 42660 GPP outputs;
Case I: building the surrogates of reduced data after SVD, and Case II: building the surrogates of
all GPPs directly. In Case I, we first apply SVD to reduce the training data dimensionality, then
build surrogates of the singular value coefficients, and last transfer the surrogate system back to
the original QoIs (i.e., the 42660 GPP variables).
The goal of this study is to develop a surrogate method that builds an accurate surrogate
system with small training data, so as to reduce the computational costs in simulating the
expensive ESMs. To demonstrate the effectiveness and efficiency of our method, we compare
the surrogate performance of the two cases in predicting the 1000 test data from two aspects: (1)
for the same number of training data, the predictive accuracy of the two surrogates, and (2) the
number of training data used to achieve the similar predictive accuracy.
Figure 4 shows the singular value decay of decomposition of the training data matrix
having 20 samples and 42660 GPP variables. The figure indicates that the singular values decay
very fast. The first 2 singular values drop about 1 magnitude, and the first 5 singular values can
capture 97% information of the training data matrix. To choose a suitable number of singular
value coefficients (Nsvd) to compress the training data and build a surrogate for, we consider a
series of Nsvds, where Nsvd=1, 5, 10, 15, and 20, and investigate their impact on NN
performance. To make a fair comparison, the same NN architectures are used for all Nsvd cases.
We consider a simple NN with 2 hidden layers and each hidden layer has 10 nodes. Figure 5
shows the prediction performance of the NNs based on the 20 training data. The figure indicates





that with considering only 1 singular value coefficient, the averaged MSE of the predictions is
about 0.053, and the NN model can fit the sELM simulation data well with the $R^2$ score of 0.83.
When 5 singular value coefficients are considered, the NN prediction accuracy improves with the
MSE of 0.02 and the $R^2$ score of 0.93. After Nsvd=5, the MSE and $R^2$ score have minor changes,
suggesting that for the limited 20 training data, Nsvd=5 is a good choice to compress the GPPs
and build a surrogate for. At this time, the surrogate error becomes dominant compared to the
SVD approximation error and including more than 5 singular value coefficients would barely
improve the NN prediction unless more training data are included to reduce the surrogate error.
In the following, we consider Nsvd=5 in Case I and compare its surrogate prediction
performance with Case II which builds surrogates for all GPPs directly.

In Case I, our method is able to use 20 training data to build a highly accurate surrogate of

42660 GPP variables with a small MSE of 0.02 and a high $R^2$ score of 0.93. The detailed NN
performance is explained in Figure 6(a) where the training and validation loss decays in building
the surrogates of the 5 singular value coefficients are plotted. The figure indicates that the loss
functions of the two data sets have similar decay, decreasing dramatically at the first 10 epochs
and then slowly decreasing to the end of training. The closely overlapped two lines in Figure 6(a)
suggest that the trained NN captures the relationship between sELM inputs and outputs pretty
well and can give reasonable predictions of GPPs for a given parameter sample.

To make a fair comparison, we use the same NN architecture in Case II as in Case I except

that the output layer of NN in Case II has all the 42660 GPPs and the output layer in Case I has
only 5 singular value coefficients. Figure 6(b) indicates that the simple NN with 20 hidden nodes
is not sophisticated enough to capture the complex relationship between the 8 inputs and 42660
outputs. As we can see in Figure 6(b), both training and validation losses are relatively high





suggesting an underfitting. The validation loss is always larger than the training loss suggesting
that the fitted NN does not generalize well and may result in poor performance in predicting new
data. Figure 7 shows $R^2$ scores of Case II in predicting the 1000 test data. The figure indicates
that the simple NN trained by 20 data in Case II has a very poor prediction accuracy with the $R^2$
score of only 0.05, close to a constant model's performance with a zero $R^2$ score. However, with
the same NN trained by the same 20 data, our SVD-based surrogate method can achieve a high
prediction accuracy with the $R^2$ score of 0.93. This demonstrates our method's capability in using
a few training samples to build an accurate surrogate model, greatly reducing the computational
costs in generating the expensive model simulation data.

On the other hand, the poor performance in Case II suggests that a wider and deeper NN is

needed when we consider the large outputs directly. We thus increase the nodes of each hidden
layer to 100 and use this complex NN with total 200 hidden nodes to approximate the
relationship of the 8 inputs and 42660 outputs in Case II. This complex NN blows up its
parameters (including weights and biases) to 4.3 million from 255 in Case I. To fit this wide NN
and calibrate its large parameters, 20 training data are way too small to get a reasonable fit. No
matter how we adjust the NN hyperparameters, we cannot get a stable solution in training. We
then increase the training data to 50, Figure 6(c) shows that the increased data greatly decrease
the training and validation losses and the validation loss is slightly higher than the training loss,
implying a good fit. Figure 7 indicates that the complex NN with 200 hidden nodes trained by 50
data in Case II significantly improves the prediction accuracy with the $R^2$ score of 0.73.
However, Case II's predictive performance is still worse than Case I which has the $R^2$ score of
0.93. We keep increasing the training data (Ntrain) to 100 and 200 in Case II. Figure 6(d) and (e)
indicate that the increase of training data brings the validation loss closer and closer to the




training loss making the fitted NN represent the underlying sELM better and better. Figure 7
shows that the nicely fitted NNs trained by large Ntrains lead to a high prediction accuracy. With
Ntrain=100, the $R^2$ score is about 0.89, and with Ntrain=200, the $R^2$ score is up to 0.95.
However, compared to Case I using 20 training data to get predictive $R^2$ score of 0.93, Case II
uses near 200 data to get the similar accuracy, increasing 10-fold computational costs. Note that,
each training data involves one sELM simulation and one regional sELM run takes about 24
hours on one processor. Thus, our SVD-based surrogate method greatly improves computational
efficiency in the accurate surrogate modeling.

Our method, in the means of simplifying NN architecture through data compression, not

only reduces the training data but also decreases the training time. Using 20 data to train a simple
NN with 255 parameters, our method takes about 4 seconds. In comparison, the traditional
surrogate method without data compression spends a great effort in training the complex NN
with 4.3 million parameters. As shown in Figure 7, Case II takes 270 seconds to fit the NN based
on 50 training data and 967 seconds for the 200 training data, showing a linear increase in
computing time. The long training time leads to high computational costs in NN hyperparameter
optimization where massive NN training are involved in searching the wide hyperparameter
space for a high-performing NN model, as discussed in the following section 3.2.
**3.2   NN's hyperparameter optimization improves surrogate accuracy**

NN has a large number of hyperparameters. Here we adjust 5 hyperparameters and use

Case I to investigate their influence on surrogate prediction accuracy. The 5 hyperparameters are,
the number of hidden layers (L) where we consider the most 3 hidden layers, the number of
nodes in hidden layer 1 (N1), in hidden layer 2 (N2), and in hidden layer 3 (N3), and the learning
rate (lr) of Adam optimization algorithm. We consider the following choices: L={2, 3}, N1={10,



20, 40, 60, 80, 100}, N2={10, 20, 40, 60, 80, 100}, N3={0, 10, 20, 40, 60, 80, 100}, and
lr=U[0.001, 0.1]. The first four hyperparameters are discrete variables and the last one, lr, is a
continuous variable with uniform distribution. The choice of L determines the selection of N3
showing a tree-like structure. We use tree-structured Parzen estimator (TPE) to search the 5
hyperparameter space and find a set of values that gives the best-performing NN. We fix the
activation function as ReLU (Agarap, 2018) which has been widely used and shown to produce
good NN predictions.

We use TPE to evaluate 100 sets of hyperparameters and the one giving the best validation

score, i.e., the smallest MSE on validation data, is chosen as the optimal hyperparameters.
Results indicate that the combination of N1=10, N2=10, N3=0, and lr=0.08 gives the best
validation score. To investigate the impact of hyperparameters on NN prediction accuracy, we
show the 100 sets of hyperparameters and their resulting $R^2$ scores in predicting the 1000 test
data in Figure 8. The figure indicates that different hyperparameter values result in dramatically
different NN performance. The prediction $R^2$ scores range from 0.66 to 0.93 where 32
hyperparameter sets have the $R^2$ scores over 0.90. The selected optimal NN producing the
smallest MSE on the validation data also gives the best prediction performance on the test data
with the $R^2$ score of 0.93. It is desired that the best NN model chosen by validation data gives the
best predictions, however, in practice it is not always the case, especially when the prediction
data deviates a lot from the validation data. Extrapolation is always a difficulty in surrogate
modeling and several researches are going on to improve the extrapolation accuracy (Gal, 2014).

Although NNs perform significantly different with different combination of

hyperparameters, the TPE algorithm can efficiently find the high-performing NNs based on
previous samples information. As shown in Figure 8, good-performing NNs prefer simple





architectures with 2 hidden layers, e.g., most blue lines have N3 of 0. After TPE finds a good
architecture of N1=10 and N2=10, it samples around this architecture in the hyperparameter
space to fine tune the learning rate till finds the most suitable lr of 0.08. This work considers 5
hyperparameters with limited choices, increasing the dimensions and possible choices of the
hyperparameters would make the search more thorough and could produce a better-performing
NN. Our surrogate method with SVD can accelerate the optimization process by reducing the
NN training time.

### 3.3     Evaluation of surrogate accuracy on large-scale spatial and temporal data

We, using only 20 expensive sELM runs, fast build an accurate surrogate system of 42660
GPPs at 1422 locations for 30 years. Therefore, for a data-model integration problem with the
QoIs within the spatial and temporal ranges, we can directly extract the information of interest
from the surrogate system to advance the analysis. The best-performing NN generated from our
method gives an overall accurate prediction of the 42660 GPPs with averaged MSE of 0.02 and
$R^2$ scores of 0.93. When using the subset of the surrogate system for data-model integration
studies, it is desired to analyze the surrogate accuracy at individual locations for specific times.
Figure 9 shows averaged $R^2$ scores over 30 years at 1422 locations. The figure indicates
that the surrogate accuracy is not uniformly good for all the locations. We observe that most
locations have $R^2$ scores above 0.9 with the best $R^2$ score of 0.96, and about 100 locations have
$R^2$ scores below 0.90 with the smallest $R^2$ score of 0.79. We highlight the locations having zero
GPP simulations in blue circles and find that these locations generally have poor predictions with
low $R^2$ scores. Connecting to Figure 2 where we label the locations in column-wise from south to
north and from west to east, we identify that those locations with zero GPPs are mostly located in



the north where the temperature is relatively low and annual GPPs tend to be zero for parameter
samples.

We pick 3 locations to closely evaluate the surrogate accuracy (Figure 9). Location 1046

has the best prediction with the highest $R^2$ score, location 1345 has the worst prediction
accuracy, and location 428 performs best among the locations with zero GPP simulations. Figure
10 shows annual GPP simulations based on sELM and NN-based surrogate in evaluating the
1000 test data for 30 years at the 3 locations. It can be seen that NN has difficulty in fitting zero
GPP data. At location 1046 where the annual GPPs are relatively high with positive values, NN
produces a great fit with a high $R^2$ score of 0.96 and a small MSE of 0.013. Location 1046
(Figure 2) is close to the lake where the variance in atmospheric drivers (e.g., temperature) is
moderated. This reduced variance leads to a smooth response surface of GPP for which NN can
easily build an accurate surrogate. In contrast, location 1345 has a large number of simulated
GPPs less than 1.0 including many zero GPPs. NN shows difficulty in predicting these small
GPPs resulting in a relatively poor performance with the $R^2$ score of 0.79. Location 1345 is
sitting in the north and has the lowest mean annual temperature, so the most parameter samples
cause low vegetation growth and small GPP values. Moreover, location 1345 is far away from
the lakes and has a large variation in atmospheric drivers. Since this location has a climate that is
at the extreme end of the range for deciduous forests, the model response is expected and
reasonable. However, this leads to a strong nonlinear response surface that casts difficulty in
surrogate modeling. In comparison, although location 428 is located in the north with some small
GPPs including zero values, it is also close to the lake which has a small variance in the
atmospheric drivers. Thus, the NN prediction performance in location 428 is not bad with the $R^2$
score of 0.91.





Figure 11 plots the averaged $R^2$ scores over all locations for 30 years. The $R^2$ scores have

small fluctuations between 0.93 and 0.94, displaying a uniformly good fit among the simulated

547       years. So, when using the surrogate model at any specific year for a data-model analysis, we

should be able to obtain a good approximation. In this study, we are considering annual GPPs.

Although the variation of atmospheric drivers between years has an impact on surrogate

accuracy, its influence is less strong compared to monthly GPPs, so a uniformly good fit among

551       years is expected.

Building a surrogate of the discontinuous response surface, e.g., vegetation turns from

alive to dead representing as the GPP jumps from nonzero to zero, is a difficulty for almost all

the state-of-the-art surrogate methods. Nevertheless, NNs, attribute to the layered architecture

and the nonlinear activation function, usually show better performance compared to other

surrogate approaches. To improve the surrogate accuracy for strong nonlinear and discontinuous

problems, one strategy is using physics-informed domain decomposition methods to build

surrogate models separately in different response surface regimes. This strategy requires the

surrogate methods strongly connecting to the simulation model, and the methods are generally

problem-specific requiring experts' interaction. Another strategy is increasing the training data to

explore complex problems. This strategy requires an increase in computational costs for extra

expensive model simulations. In the following section 4, we investigate these two strategies and

discuss their influence on surrogate accuracy.

**4    Discussion**

ESMs are complex whose response surfaces always display strong nonlinearity and

discontinuity, casting a challenge to surrogate modeling. In this section, we consider the

strategies of physics-informed learning and increase of training data to improve the surrogate





accuracy. We conduct two corresponding experiments to investigate our method's performance
in application of these two strategies. In experiment I, we divide the parameter space into two
parts producing zero GPPs and nonzero GPPs, and we use 20 training data to build surrogates of
the 42660 GPPs in the regime generating nonzero GPP samples. In experiment II, we build the
surrogates of the 42660 GPPs in the original parameter domain (Figure 3), but with increasing
training data of 200 and 1000.

We use the results of Case I as a baseline to investigate our method's performance in the

two experiments. Figure 12 shows averaged $R^2$ scores over 30 years at the 1422 locations in
experiment I. The figure indicates that without zero GPPs our method can produce a very
accurate surrogate at all locations with a uniformly high $R^2$ score of 0.98. Building the surrogates
in the subdomain without zero GPPs not only significantly improves the prediction accuracy in
locations originally having poor fit in Case I, but also further improves the prediction accuracy in
locations which already have a good fit in Case I. For example, the $R^2$ score is dramatically
improved from 0.79 to 0.97 at location 1345, from 0.96 to 0.99 at location 1046, and from 0.91
to 0.98 at location 428. As shown in Figure 13, the NN almost perfectly reproduces sELM
simulations at these 3 locations. Experiment I indicates that physics-informed domain
decomposition can be a good strategy to improve surrogate accuracy. For smooth problems (e.g.,
no sharp jumps from non-zeros to zeros in response surfaces), our method can build a very
accurate surrogate model based on a few training data.

Figure 14 shows averaged $R^2$ scores over 30 years at 1422 locations based on 200 and

1000 training data in experiment II. The figure indicates that an increase of training data greatly
enhances NN prediction accuracy. Adding 10 folds additional data from Ntrain=20 to
Ntrain=200, the overall $R^2$ score improves from 0.93 to 0.98; further increasing Ntrain to 1000,





the averaged $R^2$ score is up to 0.993 with the worst value of 0.96. Although we observe similar
nonuniform performance among locations in Figure 14 as in Figure 9, where the locations with
zero GPPs have smaller $R^2$ scores than others, increasing Ntrain significantly improves the
accuracy at all locations, especially those originally having poor fits in Case I. For example,
when Ntrain=200, most blue-circled locations have $R^2$ scores above 0.95 and for Ntrain=1000,
the $R^2$ scores at these blue-circled locations are above 0.985 in comparison to the values below
0.9 when Ntrain=20. In the examination of the 3 individual locations by comparing Figure 10
and Figure 15, we see that at the location of 1046, an increase of Ntrain enables the NN to
perfectly predict sELM simulations with negligible MSEs. Even for the location 428 with zero
GPPs, more training data can capture the discontinuous behavior better with $R^2$ score of 0.99 and
MSE of 0.003 when Ntrain=1000. The worst location happens at 1345 for all cases due to its
highly changed atmospheric drivers. Even so, the increase of Ntrain can still dramatically
enhance the NN's capability in simulating the difficult response surface. Experiment II indicates
that increasing training data is able to significantly improve the surrogate accuracy. Our method
scales well with the increase of training data and greatly improves prediction accuracy as Ntrain
increases.

The analysis of the two experiments suggests that our method is data-efficient for

continuous problems. To improve the surrogate accuracy in discontinuous and highly nonlinear
problems, we can use the physical-informed domain decomposition to focus on the continuous
and smooth regions of the response surface. If the discontinuity is the inherent feature of the
underlying function that we need to surrogate, an increase of training data would be a good
solution for surrogate accuracy improvement.



Having built a surrogate system of many GPP variables over large spatial and temporal
domains provides great flexibility and possibility for subsequent predictive analytics tasks. For
example, the surrogate model can be used for analyzing sensitivities of model parameters to any
set of spatial and temporal GPP variables, and for parameter optimization and uncertainty
quantification based on a single-site or multiple-site, a single-year or multiple-year GPP
observations using any defined objective functions. In addition, with the newly collected
observations from additional sites or further time periods, we can use the same surrogate system
for analysis as long as the QoIs are within the surrogate simulation ranges. In the future study,
we will pursue the data-model integration using the constructed surrogate system.
**5     Conclusions**
In this work, we develop an SVD-enhanced, Bayesian-optimized, and NN-based surrogate
method to improve the computational efficiency of large-scale surrogate modeling, so as to
advance model-data integration studies in Earth system model simulations. Our method is data
efficient in the fact that only 20 model simulations are needed to build an accurate surrogate
system. This is a promising result because large Earth system model ensembles are always
computationally infeasible, and 20 is a reasonable and affordable number of simulations to
consider. In addition, our method is general purpose and can be efficiently applied to a wide
range of Earth system problems with different spatial scales (local, regional, or global) at
different simulation periods. It is super effective for smooth problems and scaled well for highly
nonlinear and discontinuous problems.
We apply our surrogate method to a regional ecosystem model. The results indicate that
using only 20 model runs, we can build an accurate surrogate system of 42660 spatially- and
temporally-varied GPPs with the $R^2$ score of 0.93 and MSE of 0.02. For locations with robust





vegetation growth across the ensemble, our method can almost perfectly predict the model
simulations with the $R^2$ score of 0.96. For locations with low vegetation growth for some
parameter samples and large variation in atmospheric drivers that cause discontinuous response
surfaces, using physics-informed domain decomposition or the increase of training samples, our
method can produce accurate predictions with the $R^2$ score of 0.97 and 0.96, respectively. This
application demonstrates our method's capability in accurately reproducing expensive model
simulations based on a few parallel model runs.
**Data availability**
All the data used in this study are model simulation data, which can be generated by running
the sELM.
**Code availability**
sELM is presented in its 1.0 version, which is realized in the Python language. It is an open-
use    computer    code    which    can    be    accessed    freely    from
https://github.com/dmricciuto/OSCM_SciDAC/tree/master/models/simple_ELM.    The    source
code of surrogate modeling using machine learning techniques can be provided upon request via
lud1@ornl.gov.
**Author contribution**
Dan Lu developed the methods and carried them out. Daniel Ricciuto developed the model
code and performed the model simulations. Dan Lu prepared the manuscript with contributions
from all coauthors.
**Acknowledgments**
Primary support for this work was provided by the Scientific Discovery through Advanced
Computing (SciDAC) program, funded by the U.S. Department of Energy (DOE), Office of





Advanced Scientific Computing Research (ASCR) and Office of Biological and Environmental
Research (BER). Additional support was provided by BER's Terrestrial Ecosystem Science
Scientific Focus Area (TES-SFA) project. The authors are supported by Oak Ridge National
Laboratory, which is supported by the DOE under Contract DE-AC05-00OR22725.

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




**List of Figures**


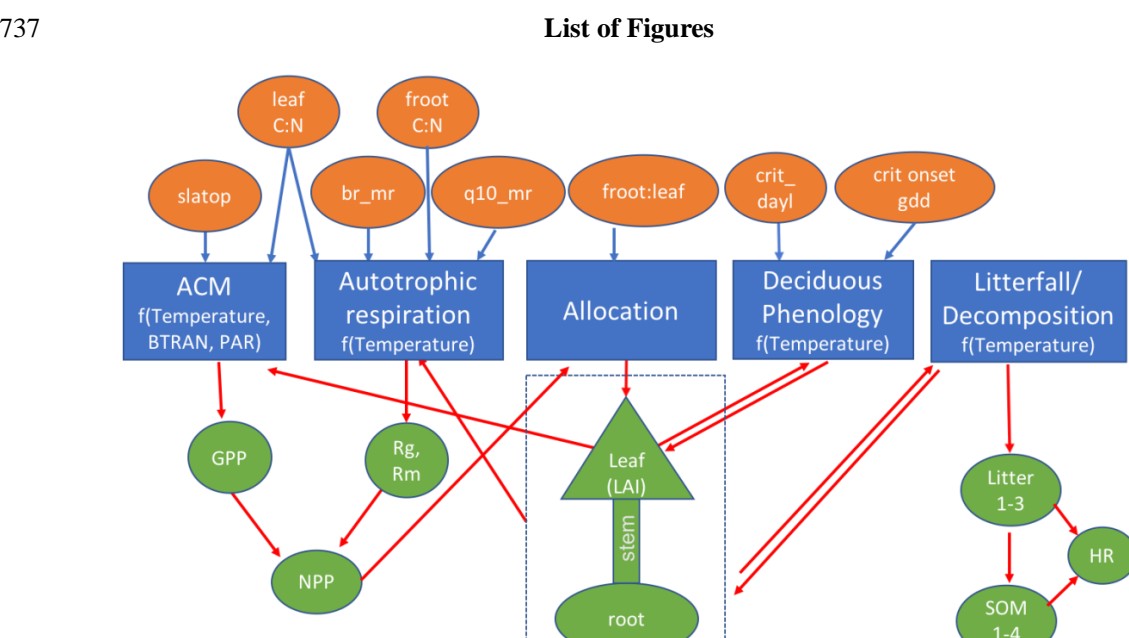


Figure 1. Schematic of sELM, where processes are shown using blue boxes with dependencies
on environmental data, 8 uncertain parameter inputs are listed in orange ovals, and model state
variables are indicated by green shapes. Parameters are input to one or more processes as
indicated by blue arrows.  Model state variables may be outputs for some processes and input for
other processes as indicated by red arrows.






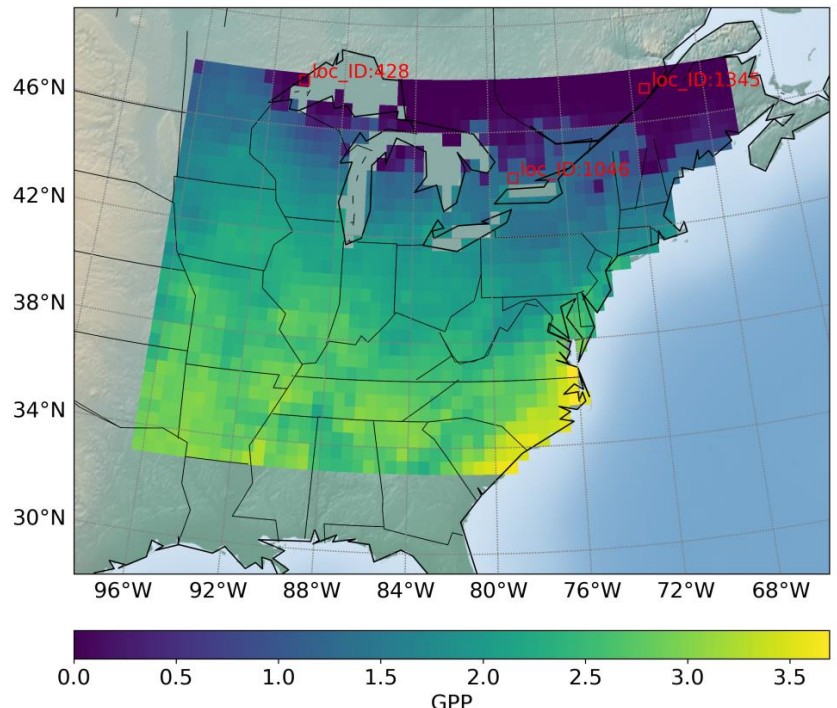


Figure 2. Locations of interest for which we build surrogates of GPP (gC/m^2/day) variables;
total 1422 locations are considered. The figure shows the sELM simulated annual GPP based on
one parameter sample.





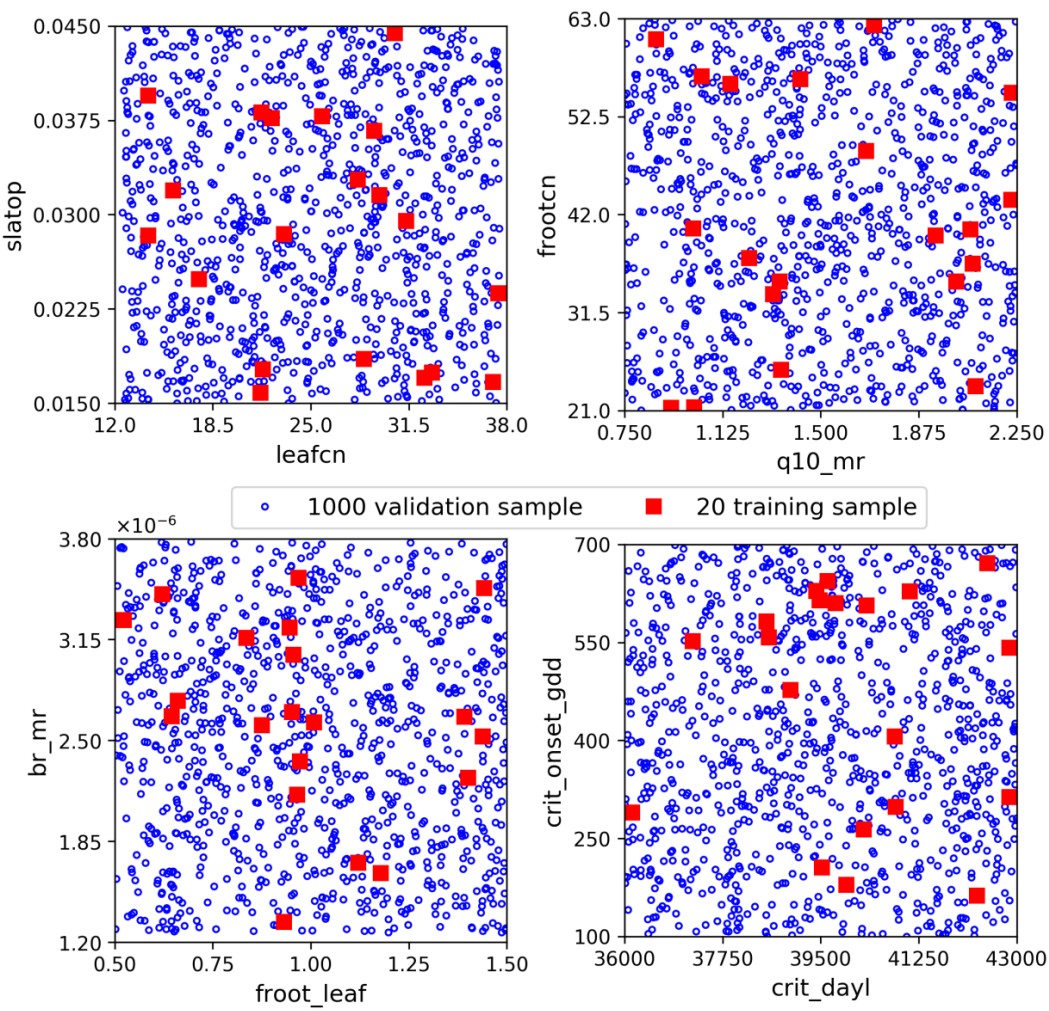

Figure 3. We consider 8 uncertain parameter inputs whose ranges are shown as axis limits. The
20 training and 1000 test data are randomly drawn from the parameter space.





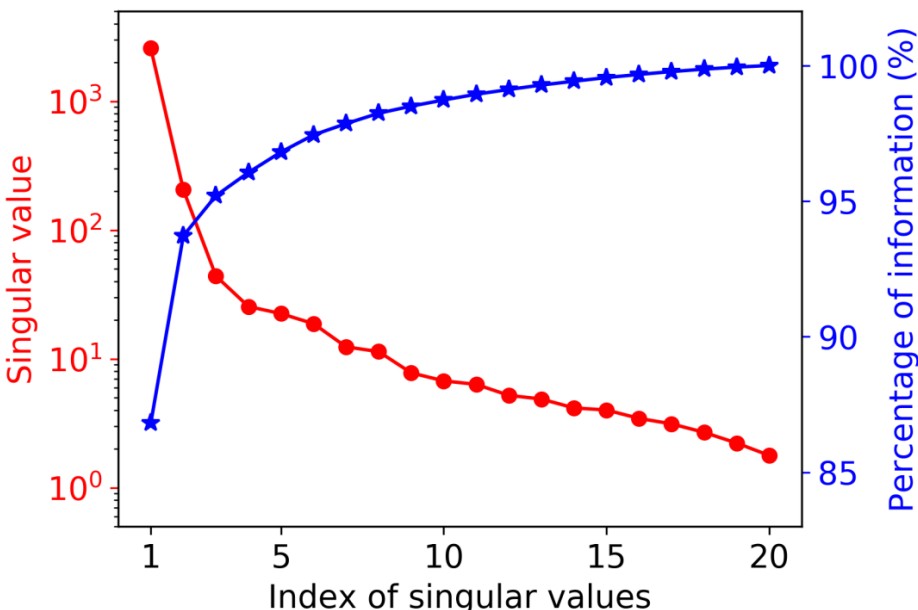


Figure 4. Singular value decay and the information contained in the first largest singular values.
The top 5 singular values contain 97% information of training data matrix with 42660 GPP
variables and 20 samples.




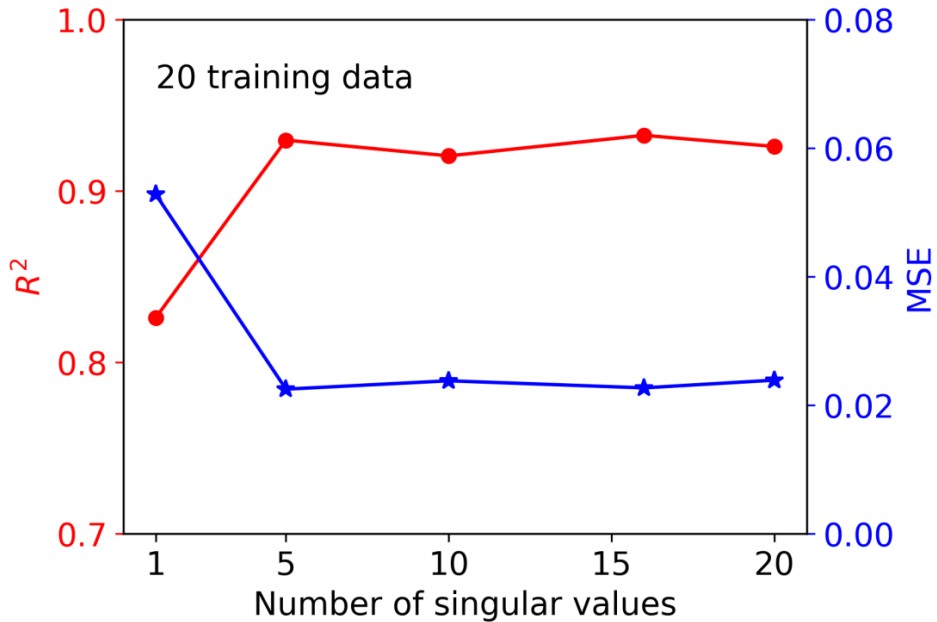


Figure 5. Performance of the NNs trained by 20 data with considering the different number of
singular value coefficients after SVD.

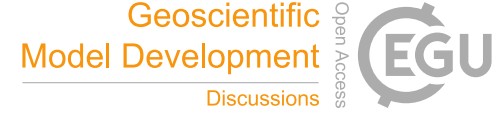

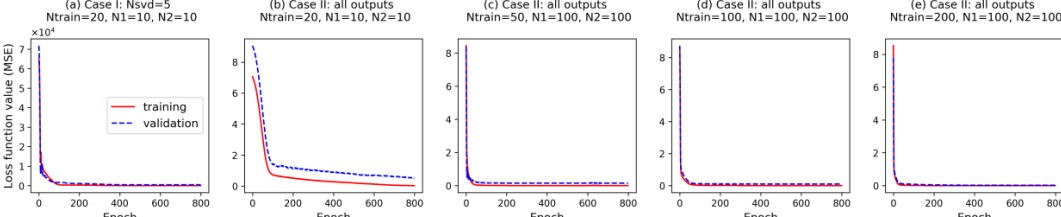


Figure 6. Changes of loss function values along epochs for training and validation data (a) in
Case I which builds surrogates of the 5 singular value coefficients with a simple NN (two hidden
layers and each layer has 10 nodes, N1=N2=10) based on 20 training data (Ntrain=20), and (b-e)
in Case II which builds surrogates of all outputs with different NN architectures and different
training data size.





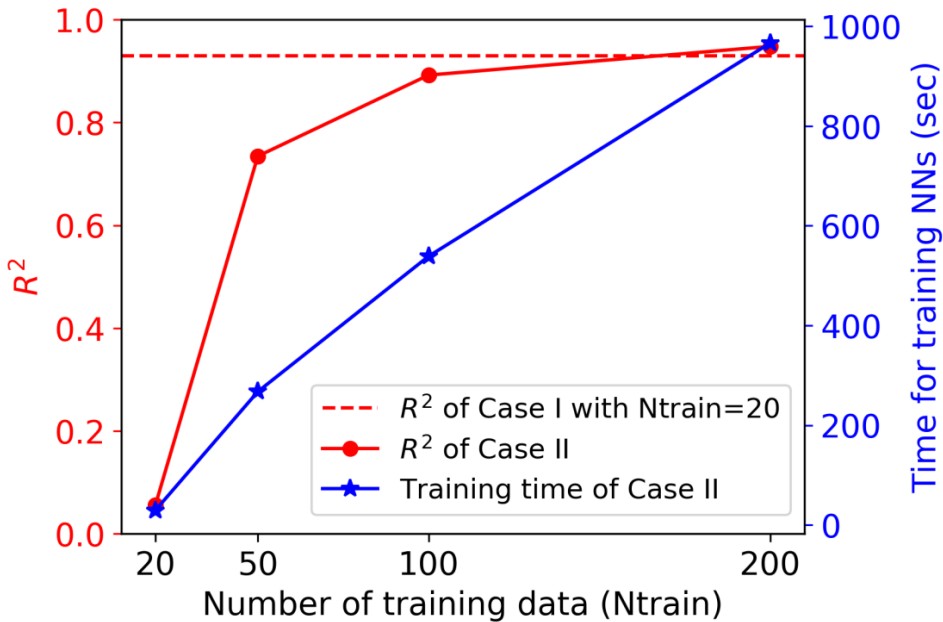


Figure 7. Comparison of NN performance between Case I: building surrogates of 5 singular
value coefficients (Nsvd=5) after SVD based on 20 training data (red dashed line) and Case II:
building surrogates for all outputs directly with different numbers of training data (red solid
line). Each training data represents one sELM simulation. The right y-axis shows the time in
training the NN in Case II. The time for training the NN in Case I is 4 seconds.



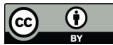

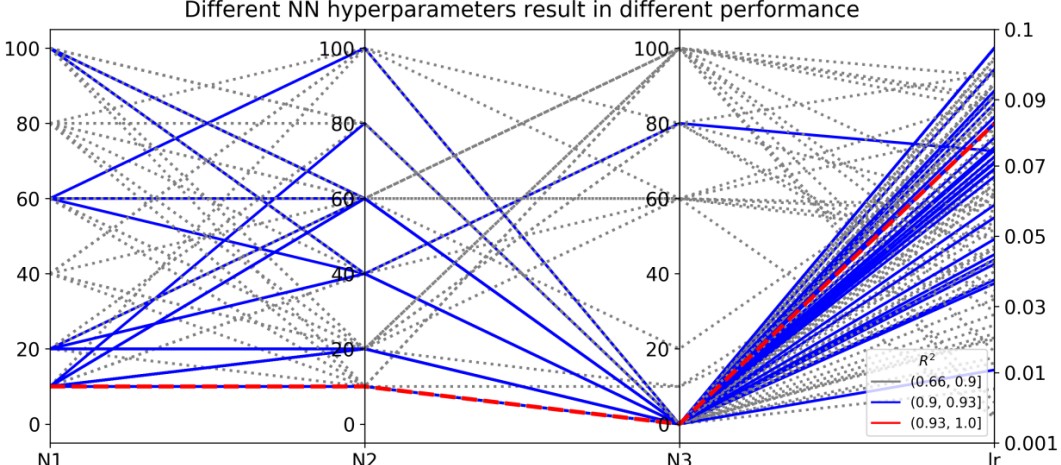


Figure 8. Different sets of NN hyperparameters result in different $R^2$ score in evaluating the 1000
test data. N$l$ is the number of nodes in hidden layer $l$, where $l$=1, 2, and 3. lr is the learning rate
of Adam algorithm for NN weights optimization.






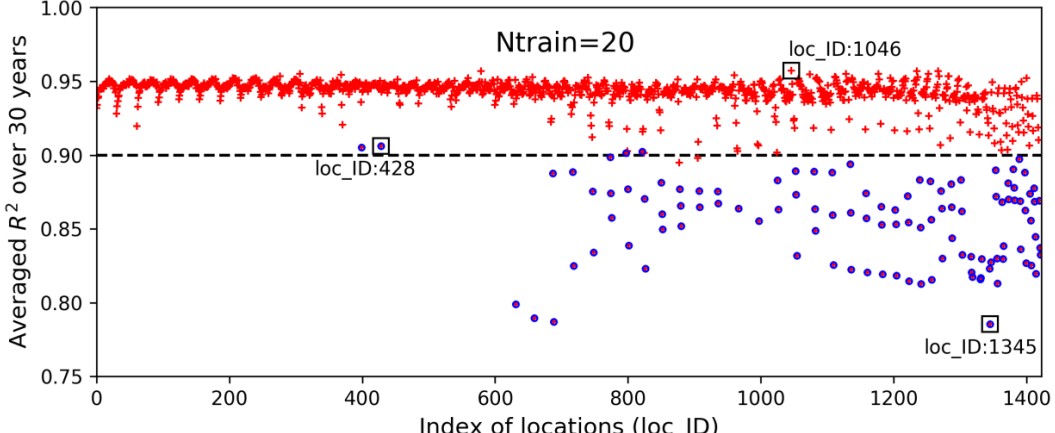


Figure 9. Averaged $R^2$ scores over 30 years at 1422 locations in evaluating the 1000 test data
based on 20 training samples, where the blue circles identify the locations having zero GPP
simulations.





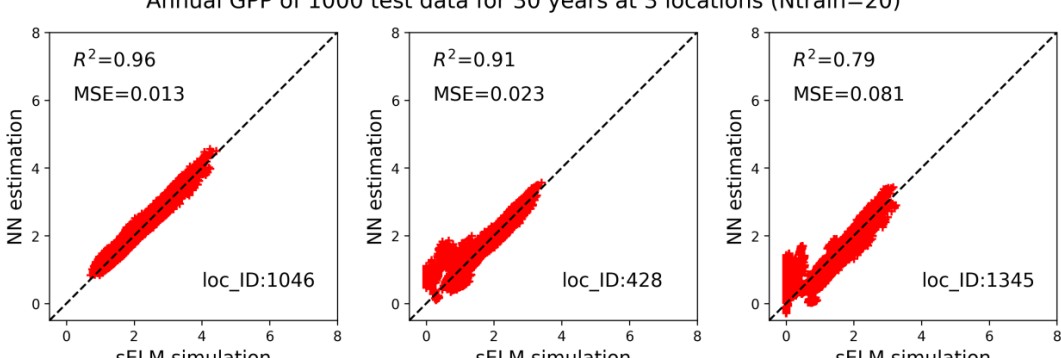


Figure 10. Simulations of annual GPPs (gC/m^2/day) from sELM and NN-based surrogate
model in evaluating 1000 test data for 30 years at 3 locations, where the NN is trained by 20 data
using our method.





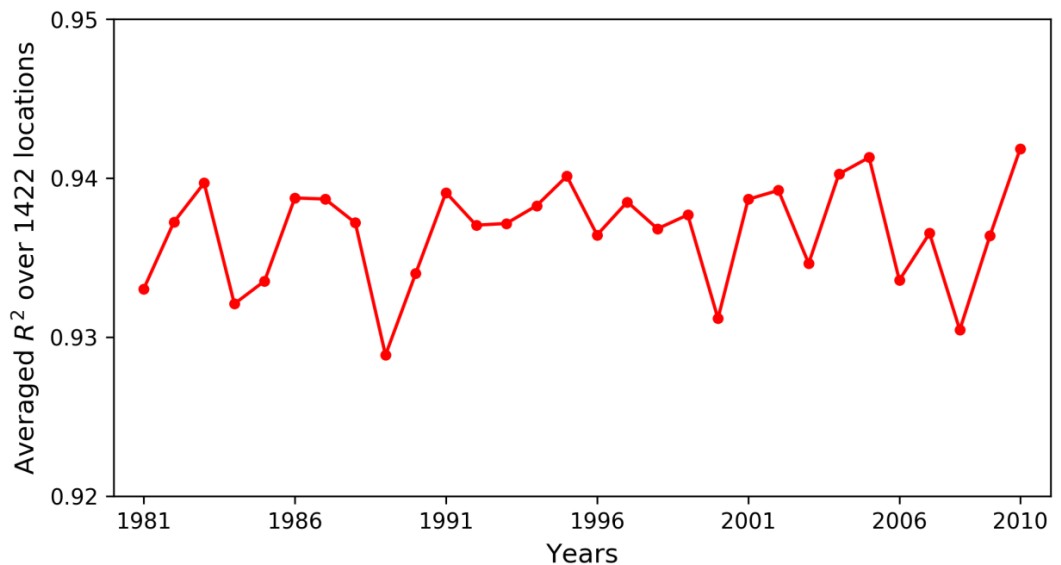


Figure 11. Averaged $R^2$ scores over 1422 locations at 30 years in evaluating the 1000 test data.






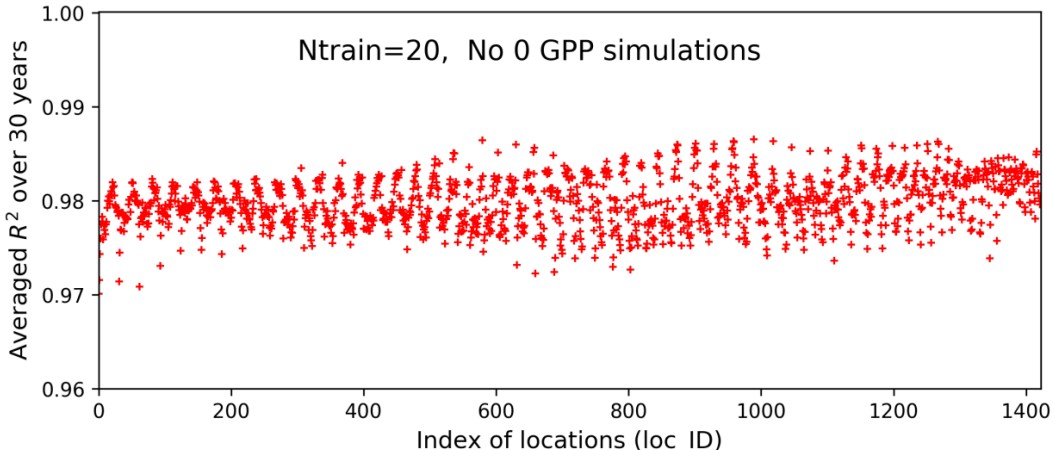

Figure 12. Averaged $R^2$ scores over 30 years at 1422 locations in evaluating the 1000 test data
based on 20 training data in experiment I where the samples are generated in a subdomain of the
parameter space without zero GPP simulations. The averaged $R^2$ score is 0.98.





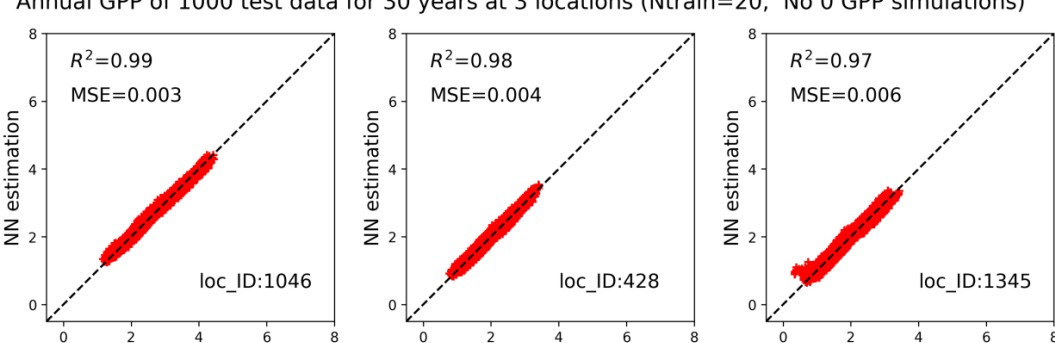

Figure 13. Simulations of annual GPPs (gC/m^2/day) from sELM and NN-based surrogate model in evaluating 1000 test data for 30 years at 3 locations in experiment I where the samples are generated in a subdomain of the parameter space without zero GPP simulations.






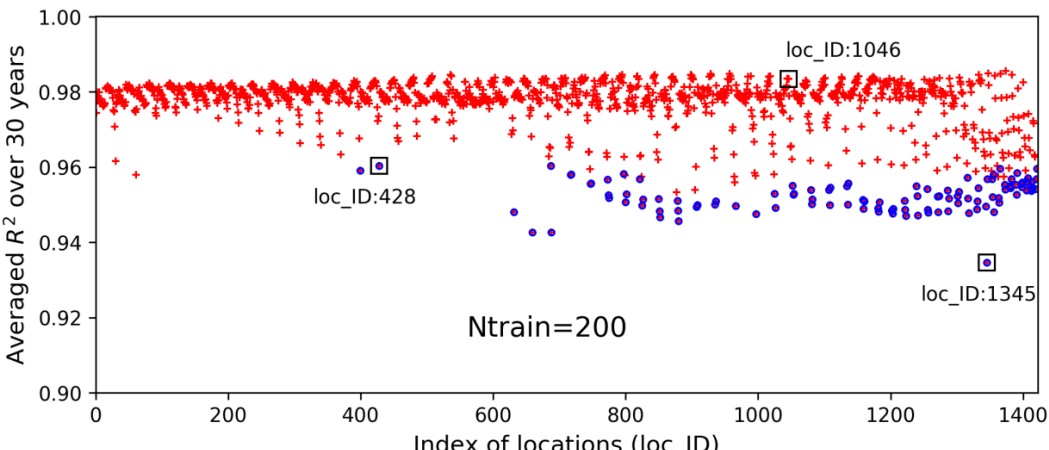


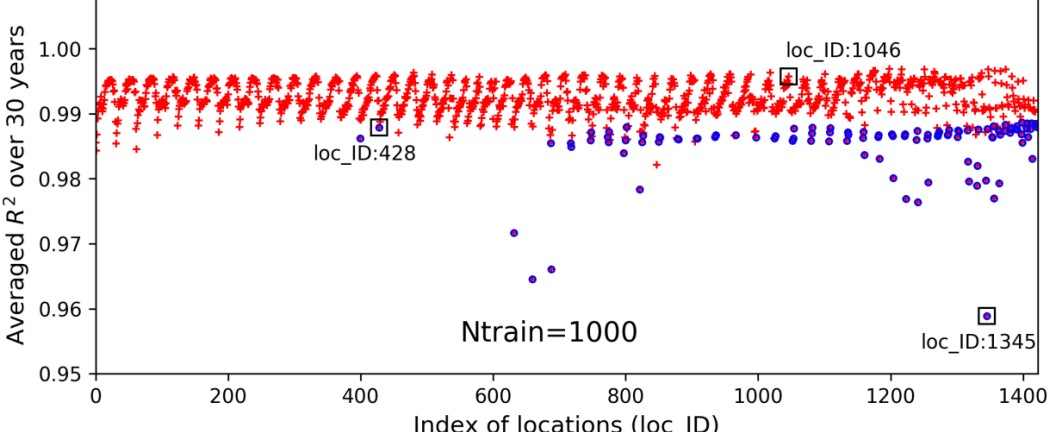

Figure 14. Averaged $R^2$ scores over 30 years at 1422 locations in evaluating the 1000 test data
based on 200 and 1000 training samples, where the blue circles identify the locations having zero
GPP simulations.





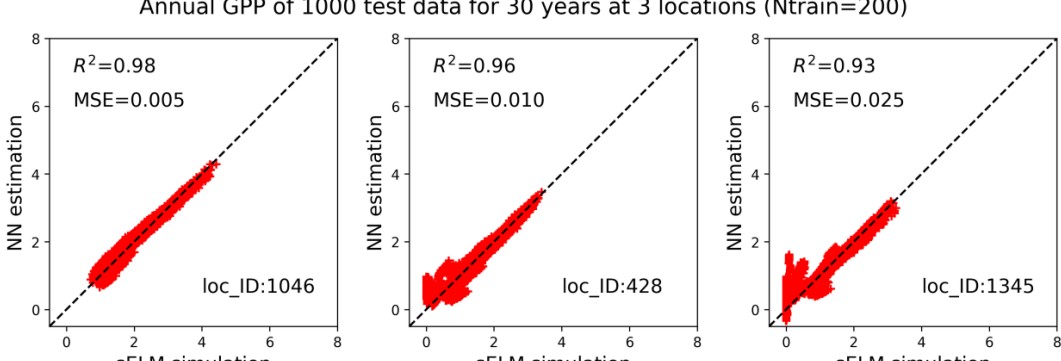


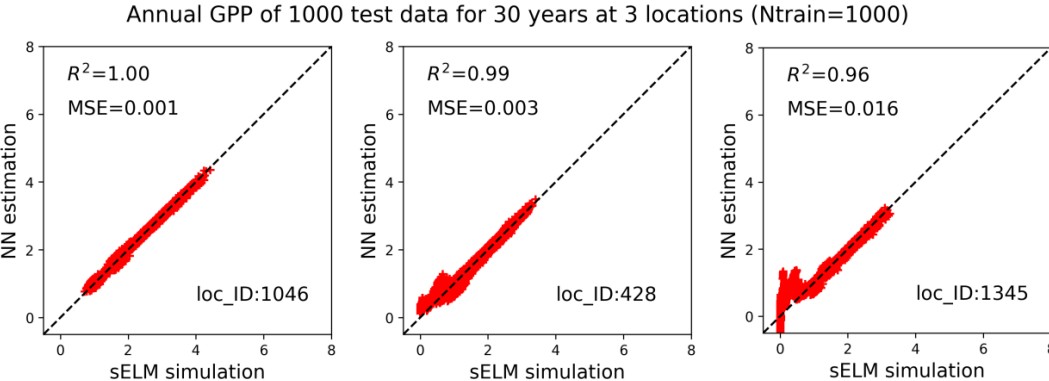

Figure 15. Simulations of annual GPPs (gC/m^2/day) from sELM and NN-baed surrogate model
in evaluating 1000 test data for 30 years at 3 locations, where the NN is trained by 200 and 1000
data.