# Peer review of "Efficient surrogate modeling methods for large-scale Earth system models based on machine learning techniques"

_Geoscientific Model Development, 2018_

## Referee Comment (RC1) · Xiankui Zeng (Referee) · 9 Feb 2019

Comments to "Efficient surrogate modeling methods for large-scale Earth system models based on machine learning techniques" submitted to *Geoscientific Model Development*.

**General Comments:**

The heavy model evaluation time is always the burden of simulation and prediction of complex earth system. This study developed a technique to build surrogate models for a large-scale Earth system model (ESM) with many output variables. This study uses singular value decomposition to reduce the output dimensions, and then use Bayesian optimization techniques to generate an accurate neural network surrogate model based on only 20 ESM simulation samples.

This research problem is of interest in ESM field. The manuscript is well organized and easy to read, the results and discussion are sufficient to support the conclusion. However, this paper may be improved after clearing following few questions, and a minor revision is recommended to this paper.

**Specific Comments:**

1. Line 367: For the 800 epochs to train NN model, are these epochs have the same 20 training data? and they are different at the choosing of the 6 validation data?

2. Personally, I want to see how complex is the ESM, it would be nice if the authors can present the response surfaces or contour maps of the ESM, such as the zero GPP zones;

3. Line 335, Please give a short description to the TPE method.

4. Line 555, May be some references are needed to support this statement "NNs, attribute to the layered architecture and the nonlinear activation function, usually show better performance compared to other surrogate approaches."

5. The plots of Figure 6 is not clear, please revise it.

---

## Referee Comment (RC2) · Tianfang Xu (Referee) · 25 Feb 2019

The manuscript by Lu and Ricciuto proposed a computationally efficient strategy to construct fast-to-evaluate surrogate of a large-scale Earth system model using neural networks (NN). The strategy uses singular value decomposition (SVD) to reduce the dimension of outputs, which are spatiotemporally correlated, and Bayesian optimization techniques to select NN hyperparameters. In this way, a surrogate model with satisfactory accuracy can be constructed with as few as 20 runs of the Earth system model.

I found the manuscript overall well written, experiments are clearly described, and results are clearly presented. I suggest minor revision based on comments listed below, focusing on improving clarity of text and adding necessary citations. I also made a suggestion regarding publishing data from this study.

1. It is not clear from the manuscript the advantage of (1) using a single surrogate model for all model outputs (the strategy taken in this manuscript) versus (2) constructing one surrogate for each model outputs (e.g., Xu et al., 2017). As mentioned in Introduction, (1) leads to very large neural network and storage and operation of large matrices. Using (2) will lead to a very large number of NN models. However, each NN can be very simple, and its evaluation time will be negligible. Which method is more computationally efficient may depend on specific case. I suggest to add some discussion about the tradeoff.

2. For the case study of 8 parameters, it was found that 20 model runs suffice. How would the needed number of model runs scale with increasing number of parameters? This could be an advantage of the proposed methods over existing methods such as gPC.

3. Would the surrogate modeling performance depend on the location of the 20 parameter sets (training data)? It will be helpful to provide some guidance on how to select parameter sets to generate training data.

4. It is argued in several places that a slight change in hyperparameters can result in dramatically different NN performance. I suggest providing a citation or two to support this statement.

5. It is not entirely clear to me how the Bayesian optimization of the hyperparameters is implemented. I suggest to include more details to help interested readers, for example, what is the prior and posterior.

6. Line 44-49: from the context data-model integration refers to calibration/uncertainty quantification. However in other context the term can also refer to data assimilation

and other methods.

7. Line 261: what does "information" refer to? Do you mean variance?

8. I wonder whether it would be possible to publish (on a data repository such as Hy-droShare, https://www.hydroshare.org/) the training and testing data generated by the sELM model, given that sELM is computationally expensive. This will provide a very interesting and representative dataset for the uncertainty quantification community. This is only a suggestion, and I leave it entirely for the authors to decide.

References Xu, T., Valocchi, A. J., Ye, M., & Liang, F. (2017). Quantifying model structural error: Efficient Bayesian calibration of a regional groundwater flow model using surrogates and a data driven error model. Water Resources Research, 53(5), 4084-4105.

---

## Author Comment (AC1) · 22 Mar 2019

**Response to Reviewer #1 Comments on Manuscript gmd-2018-327**

We would like to express our sincere gratitude to Dr. Xiankui Zeng for his insightful and constructive comments and suggestions. All comments have been addressed in the revised manuscript as highlighted in red. Below is an item-by-item response to the comments.

**General Comments:**

The heavy model evaluation time is always the burden of simulation and prediction of complex earth system. This study developed a technique to build surrogate models for a large-scale Earth system model (ESM) with many output variables. This study uses singular value decomposition to reduce the output dimensions, and then use Bayesian optimization techniques to generate an accurate neural network surrogate model based on only 20 ESM simulation samples.

This research problem is of interest in ESM field. The manuscript is well organized and easy to read, the results and discussion are sufficient to support the conclusion. However, this paper may be improved after clearing following few questions, and a minor revision is recommended to this paper.

**Response:**

We appreciate the reviewer for the positive evaluation. The comments have been addressed in detail below and corresponding revisions have been highlighted in red in the revised manuscript.

**Specific Comments:**

**Comment 1:**

Line 367: For the 800 epochs to train NN model, are these epochs have the same 20 training data? and they are different at the choosing of the 6 validation data?

**Response:**

Yes. As explained in lines 352-359 of the original manuscript, we consider 20 training data and the validation data is chosen as 0.3 fractions of the training data. In each epoch, the training data is shuffled, and the validation data is always selected from the last 0.3 fraction.

**Comment 2:**

Personally, I want to see how complex is the ESM, it would be nice if the authors can present the response surfaces or contour maps of the ESM, such as the zero GPP zones.

**Response:**

Figure 1 of the manuscript shows schematic of the sELM which includes five major processes. In this study, we consider 8 uncertain parameters. Since it is a high-dimensional problem, we are unable to visualize the GPP response surface on the entire parameter space. The following Figure 1 shows the response surface of averaged annual GPP over 30 years at location 1345 based on parameters *leafcn* and *slatop* using 1000 samples. The figure indicates that the response surface is rather rough, and the most zero GPP values are caused by large *leafcn* samples.

[Figure]

Figure 1. Response surface of averaged annual GPP over 30 years at location 1345 based on parameters *leafcn* and *slatop* using 1000 samples.

**Comment 3:**

Line 335, Please give a short description to the TPE method.

**Response:**

We thank the reviewer for the suggestion. A detailed description of the TPE method has been added in Section 2.2.3 of the revised manuscript.

**Comment 4:**

Line 555, May be some references are needed to support this statement "NNs, attribute to the layered architecture and the nonlinear activation function, usually show better performance compared to other surrogate approaches."

**Response:**

We thank the reviewer for the suggestion. References have been added in the revised manuscript.

**Comment 5:**

The plots of Figure 6 is not clear, please revise it.

**Response:**

Figure 6 has been updated for a clear presentation.

---

## Author Comment (AC2) · 22 Mar 2019

**Response to Reviewer #2 Comments on Manuscript gmd-2018-327**

We would like to express our sincere gratitude to Dr. Tianfang Xu for her insightful and constructive comments and suggestions. All comments have been addressed in the revised manuscript as highlighted in red. Below is an item-by-item response to the comments.

**General Comments:**

The manuscript by Lu and Ricciuto proposed a computationally efficient strategy to construct fast-to-evaluate surrogate of a large-scale Earth system model using neural networks (NN). The strategy uses singular value decomposition (SVD) to reduce the dimension of outputs, which are spatiotemporally correlated, and Bayesian optimization techniques to select NN hyperparameters. In this way, a surrogate model with satisfactory accuracy can be constructed with as few as 20 runs of the Earth system model.

I found the manuscript overall well written, experiments are clearly described, and results are clearly presented. I suggest minor revision based on comments listed below, focusing on improving clarity of text and adding necessary citations. I also made a suggestion regarding publishing data from this study.

**Response:**

We appreciate the reviewer for the positive evaluation. The comments have been addressed in detail below and corresponding revisions have been highlighted in red in the revised manuscript.

**Specific Comments:**

**Comment 1:**

It is not clear from the manuscript the advantage of (1) using a single surrogate model for all model outputs (the strategy taken in this manuscript) versus (2) constructing one surrogate for each model outputs (e.g., Xu et al., 2017). As mentioned in Introduction, (1) leads to very large neural network and storage and operation of large matrices. Using (2) will lead to a very large number of NN models. However, each NN can be very simple, and its evaluation time will be negligible. Which method is more computationally efficient may depend on specific case. I suggest to add some discussion about the tradeoff.

**Response:**

We thank the reviewer for the suggestion and reference. We are well aware of the reviewer's work published in Xu et al (2017). We agree with the reviewer that both are good strategies and which one should be chosen is problem specific and goal oriented. In this study, we are interested in building a surrogate system for large-scale Earth system models (ESMs). ESMs tend to be simulated in a regional or global scale

with many grid cells for several years, producing a large number of output variables. We aim to approximate the relationship between model parameters and the large spatiotemporally varied output variables. We can certainly treat these output variables individually and build a single surrogate model for each of them, but this strategy may ignore the correlation between the outputs. Also, when using the surrogates for prediction, the strategy (2) needs to evaluate all the single surrogate models.

We also agree with the reviewer that a complex neural network will be needed to approximate the relationship between the large number of inputs and outputs. So, in this study we first used SVD to reduce output dimensions and thus simplify the input-output relationship and the NN architecture.

**Comment 2:**

For the case study of 8 parameters, it was found that 20 model runs suffice. How would the needed number of model runs scale with increasing number of parameters? This could be an advantage of the proposed methods over existing methods such as gPC.

**Response:**

This is a great comment. As explained in the original manuscript, the computational complexity of gPC increases factorially fast with the parameter size and polynomial order, making gPC computationally unaffordable for large-scale complex problems with many parameters. NNs use network structure, compared to the linear combination of basis functions used by gPC, to simulate the relationship between model inputs and outputs. This network approximation enables NNs to simulate complex functional relationship but meanwhile makes its computational complexity less straightforward than the gPC. In general, the number of required training data depends more on the NN architecture than on the number of NN inputs (in this study, NN inputs are model parameters), as the training data is used to calibrate the NN parameters (i.e., weights and biases) and a simple NN architecture has a small parameter size. It is possible that a simple NN can simulate a high-dimensional problem if the problem's input-output relationship is simple, and a complex NN is needed to simulate a low-dimensional problem is the problem's input-output relationship is complex. Thus, in this study, we first used SVD to simplify the input-output relationship, so that a simple NN architecture can be appropriate and a small sample size can be sufficient to accurately train the simple NN.

**Comment 3:**

Would the surrogate modeling performance depend on the location of the 20 parameter sets (training data)? It will be helpful to provide some guidance on how to select parameter sets to generate training data.

**Response:**

In general, the training data should be diverse enough to include the testing data i.e., the untrained data) to avoid extrapolation. Extrapolation affects surrogate performance and is a challenge for almost all the surrogate modeling methods. In this study, the 20 training data and 1000 testing data are randomly selected from the parameter space as shown in Figure 3 of the manuscript.

**Comment 4:**

It is argued in several places that a slight change in hyperparameters can result in dramatically different NN performance. I suggest providing a citation or two to support this statement.

**Response:**

We thank the reviewer for the suggestion. References have been provided in the manuscript and our Figure 8 also supports the argument of the importance of hyperparameters in NN performance.

**Comment 5:**

It is not entirely clear to me how the Bayesian optimization of the hyperparameters is implemented. I suggest to include more details to help interested readers, for example, what is the prior and posterior.

**Response:**

We thank the reviewer for the suggestion. A detailed description of Bayesian algorithms for NN hyperparameter optimization has been added in Section 2.2.3 of the revised manuscript.

**Comment 6:**

Line 44-49: from the context data-model integration refers to calibration/uncertainty quantification. However, in other context the term can also refer to data assimilation and other methods.

**Response:**

We agree with the reviewer that data-model integration is a broad concept including many methods. These methods, including data assimilation approaches, are usually computationally expensive involving a large ensemble of model simulations. This work built an accurate and fast evaluated surrogate system to reduce the computing time of a single model run so as to reduce the total simulation costs.

**Comment 7:**

Line 261: what does "information" refer to? Do you mean variance?

**Response:**

Yes, the percentage of information means the percentage of the total variance explained by the singular values. This point has been clarified in the revised manuscript.

**Comment 8:**

I wonder whether it would be possible to publish (on a data repository such as HydroShare, https://www.hydroshare.org/) the training and testing data generated by the sELM model, given that sELM is computationally expensive. This will provide a very interesting and representative dataset for the uncertainty quantification community. This is only a suggestion, and I leave it entirely for the authors to decide.

**Response:**

We thank the reviewer for the suggestion. The training and testing data will be uploaded to the author's GitHub for download when the manuscript is published.

---

## Author Response (AR2)

**Response to Editor's Comments on Manuscript gmd-2018-327**

We would like to express our sincere gratitude to Dr. David Topping for his suggestions. All comments have been addressed in the revised manuscript. Below is an item-by-item response to the comments.

**Comments to the Author:**

Many thanks for taking the time to respond to all comments made. I am happy that this paper deserves a place in GMD, demonstrating new methdologies for complex simulations of large scale models using machine learning techniques. Given this has a broad remit, I encourage further studies around a family of methods.

**Response:**

We appreciate the editor for his positive evaluation. We are currently developing new machine learning methods to further improve predictive performance of terrestrial ecosystem models. We got some promising results and hopefully will submit a manuscript to GMD soon.

**Non-public comments to the Author:**

Hi guys. many thanks for responding to the reviews. I am happy for this to be published subject to a few minor corrections.

Please change 'It is an open use..' to 'open source..' in code availability. Also, on line 372 I would suggest changing 'The TPE algorithm has great improvement over the classic hyperparameter optimization' to 'The TPE algorithm exhibits significant improvement over the classic hyperparameter optimization..' Thanks!

**Response:**

We appreciate the editor for the positive evaluation. The suggested changes have been made in the revised manuscript.